# MALLM-GAN: Multi-Agent Large Language Model as Generative Adversarial Network for Synthesizing Tabular Data

## Abstract

In the era of big data, access to abundant data is crucial for driving research forward. However, such data is often inaccessible due to privacy concerns or high costs, particularly in healthcare domain. Generating synthetic (tabular) data can address this, but existing models typically require substantial amounts of data to train effectively, contradicting our objective to solve data scarcity. To address this challenge, we propose a novel framework to generate synthetic tabular data, powered by large language models (LLMs) that emulates the architecture of a Generative Adversarial Network (GAN). By incorporating data generation process as contextual information and utilizing LLM as the optimizer, our approach significantly enhance the quality of synthetic data generation in common scenarios with small sample sizes. Our experimental results on public and private datasets demonstrate that our model outperforms several state-of-art models regarding generating higher quality synthetic data for downstream tasks while keeping privacy of the real data.

## 1 Introduction

Tabular data is the most common data format in high-stakes sectors like healthcare. There are many fundamental problems in dealing with tabular data, such as data scarcity, missing values, and irregularity. Among them, the data scarcity problem has been the main roadblock. Many datasets in healthcare, such as clinical trial data, have small data sizes due to data collection costs and privacy risks, and consequently, these data cannot afford modern machine learning (e.g., deep learning), which generally has thousands of parameters, at minimum.

Recent advancements in generative models, particularly in text and image,(7; 28) have shown the benefits of technology for generating synthetic data that resembles real data. Despite this potential, generating tabular data has not been fully tapped into; it has evolved through traditional statistical approaches, like Bayesian networks (38), to deep learning techniques, including autoencoders and Generative Adversarial Networks (GANs) (42). However, these methods require large amounts of data for their training, contradicting our objective of solving data scarcity. Also, the sparsity and heterogeneity in tabular data make GAN or other deep learning a suboptimal choice, as evidenced by the fact that the tree-based method (e.g., XGBoost) works better than deep learning model (11).

Recently, advancements in large language models (LLMs) have also enabled researchers to use their general intelligence to synthesize tabular data.(6; 14) The premise is that prior knowledge encoded in the parameters of LLMs can provide contextual knowledge for coherent semantics that is required to learn the underlying data generation process. Several studies transformed tabular data to natural language via serialization, and used pre-trained LLMs to generate text containing the synthetic tabular data (6; 14; 21). While fine-tuning LLMs has led to the creation of more nuanced synthetic data, this process again requires a significant sample size, contradicting the objective of addressing data scarcity.

In contrast, in-context learning presents a promising alternative by allowing LLMs to customize without compromising their general reasoning abilities (24) . Particularly, few-shot learning in in-context learning is to provide a few "examples" of data to let LLM to learn the patterns and mimic the examples (17). Our study aims to utilize this few-shot capability for synthetic tabular data generation. Currently, tabular generative model by in-context learning, such as (31) faces a critical shortcoming;

it only accommodate too few "examples" (not an entire dataset available), thus discarding remaining data that cannot manage to fit into limited context length (input token). For example, in our real-world trial data ATACH2 with 37 variables, just ten samples consume 4,232 input tokens, whereas the common input token size of LLMs, such as GPT3, is 2,048. This failure to utilize all available scarce data can make LLMs perceive the underlying data-generating process merely based on "educated guess" with its prior knowledge instead of the data itself. This, consequently, causes distribution discrepancy between real data and synthetic data.

Therefore, we aim to bridge this critical gap. Our key idea is to make the *data generation process* explicit; the objective of our in-context learning is to generate a better data generation process, as well as to generate individual data instances. Here, the data generation process is a prompt text that consists of the context of data and any simple "model" that describes the relationship between data variables. We chose to use a Bayesian network or causal structure due to its interpretability and simplicity.

However, another challenge is to identify the ground-truth data generation process. Motivated by GAN's adversarial training, we optimize the data generation process ("generator") in adversarial training with "discriminator" (Table 1). The discriminator's role is to discriminate real data from the generated data, and we use the accuracy of the discriminator as a "loss function" to be minimized to optimize the generator. Unlike GAN, our generator is a text format, which doesn't have derivatives. We address it by Optimization by Prompting, which leverages an independent LLM as an optimizer (43). After optimizing the data generation process, the LLM as generator uses it to finally generate synthetic data.

Table 1: Comparison of Generative Adversarial Network (GAN) and Our Model

|  | GAN | Our Model |
| --- | --- | --- |
| **Generator** | Neural network | Frozen LLM and prompt |
| **Discriminator** | Neural network | Tabular data classifier |
| **Optimizer** | Gradient descent | Frozen LLM and prompt |

Contribution of this paper can be summarized as below

- *Novelty*: We propose a novel concept of optimizing data generation process using in-context learning of LLM. This leverages both data-driven supervised model (discriminator) and knowledge-driven in-context learning of LLM (generator, optimizer).

- *Few-shot synthetic data generation*: Our model works when there is too little data to train a parametric model. It mitigates data scarcity problem in many small-size tabular data in healthcare.

- *Conditional sampling*: Our generator is based on LLM, which enable conditional sampling seamlessly by prompting.

- *Explainability*: Our LLM-based generator explicitly reveals data generation process and its reasoning, which are explainable by design. This enables transparency of our model and facilitates human feedback, such as refining the knowledge.

## 2 RELATED WORK

This section provides a brief overview of the most relevant prior work in the field of tabular data generation, LLM-supported synthetic data generation, and the different roles that LLM plays in real-world applications.

**Synthetic tabular data generation**. Many studies have been proposed to generate high-quality tabular data for privacy-preserving data sharing and augment training data size for machine learning models. Traditional method including Bayesian network (44; 38), approximation Bayesian computation (5), and SMOTE (8). Particularly, a Bayesian network can represent a pairwise causal structure via directed acyclic graph (DAG) (26), in which a directed edge between variable A and B exists if A causes B. Causal structure is a compact representation of underlying variable relationship, but does not fully capture the nonlinear numerical relationship between the different types of variables. Deep

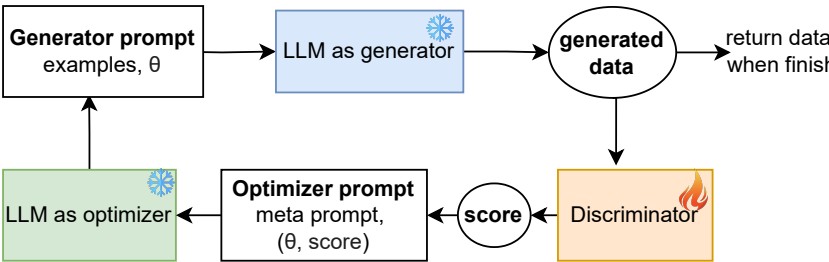

Figure 1: Overview. In each optimization step, the LLM as Optimizer generates a data generation process $\theta$ in (**Generator prompt**) based on the pairs of previous $\theta$ and its score in **Optimizer prompt**. Then the LLM as Generator use the current $\theta$ and a few examples to generate data. We evaluate the $\theta$ using the accuracy (**score**) of Discriminator . The more the data generation process $\theta$ is optimized, the lower the discriminator's accuracy. This adversarial optimization finishes when data generation process is no more improved.

generative models have also been widely utilized. This includes variational autoencoders such as TVAE (42) and others (2; 40); GAN such as CTGAN (42), and diffusion model such as TabDDPM (19). The limitation of these models is that they require a sufficient sample size to train the generative model, which contradicts our aim to solve data scarcity.

**Synthetic data generation using LLMs.** Recently, large language models emerged and have demonstrated their powerful performance in generating natural text. It has also shown a great potential in tabular data (10), such as for predicting (14; 12; 45; 21) and generating tabular data (6; 32; 47; 12). Among them, GReaT (6), the first model in this line, transformed the tabular data into text and fine-tuning LLM (GPT-2), including a feature order permutation step for added realism. However, these prior methods require fine-tuning LLMs, which requires large data size and computing resources for fine-tuning. This limitation of existing models motivates us to develop *few-shot* tabular data generative model.

**Multiple roles of LLM in real-world applications.** In addition to typical natural text generation tasks it was originally trained for, LLMs have been utilized in various tasks. LLM has been used as an optimizer for the data type that we can not calculate derivatives directly (43). This LLM as optimizer was used to optimize prompt (43) and optimize heuristic algorithms written in codes (29). LLMs also have been utilized for communicating with other LLMs (i.e., multi-agent LLM); multiple LLMs play different roles to accomplish a task collaboratively, such as coding,(13) question answering,(41) and online decision making.(35; 16)

**LLM and causal structure discovery** Causal structure discovery involves either data-driven or expert knowledge-driven approaches. Data-driven approaches discover causal structure from data by conditional independence tests (33; 34), score-based heuristics (36), or relaxing the discrete constraints into continuous optimization (48; 46). Despite these advances, identifying the ground-truth causal structure from data remains a significant challenge, particularly in complicated domains like healthcare or when data is scarce. Expert-driven approaches can be an alternative option, but these methods are time-consuming and require significant expert involvement. Recently, many studies (18; 23) suggest that LLMs, which encode prior knowledge in their parameters, can support causal discovery by complementing expert knowledge. In this paper, we leverage multiple LLMs with different roles to mimic adversarial training in GAN and use the heuristic causal structure discovery to guide the data generation process.

## 3 METHODOLOGY

### 3.1 PROBLEM FORMULATION

Given a small labeled tabular dataset with $n$ instances and $d$ features, denoted as $D_{\text{real}} = (\mathbf{x}, y)$ where $\mathbf{x}$ represents a $d$-dimensional vector of features and $y$ indicates label. The features are described by natural-language strings like "age" or "gender". For synthetic data generation, we train a generator on a training subset $D_{train}$ of $D_{\text{real}}$, generating synthetic dataset $D_{syn}$.

### 3.2 MULTI-AGENT LLM AS GAN

**Overview.** We propose to develop a multi-agent LLM as GAN (`MALLM-GAN`) that generates tabular data by mimicking adversarial optimization (Fig. 1). The objective is to optimize the data generation process $\theta$, which is a natural language description of i) the problem description and ii) the simple data generation process or causal structures representing relationships between variables. In `MALLM-GAN`, for each iteration $i$, an LLM agent **Generator** generates data $D_{syn}$ with $\theta_i$ and a batch in $D_{train}$; a supervised model **Discriminator** is accordingly optimized using $[D_{train}, D_{syn}]$ and evaluates $\theta_i$ using $D_{test}$; and another LLM agent **Optimizer** improves $\theta_i$ to decrease the discriminator's accuracy (Algorithm 1). We repeat the iterations until the discriminator's accuracy converges or the iteration reaches the maximum epoch.

#### 3.2.1 GENERATOR

**Data generation process.** The data generation process $\theta$ is described in natural language and prompts the generator LLM to create synthetic data. It includes: i) context of data collection, ii) data schema, iii) causal structure describing relationships between variables, and iv) task instruction. The context provides external knowledge on data collection (e.g., *"this dataset includes subject's socioeconomic factors..."*). The data schema contains the meta-information of variables (e.g., name, description, type, and categorical values). These elements remain constant during optimization. The causal structure, represented as a DAG and converted into text format $(x_1, x_2)$, indicates $x_1$ causes $x_2$. Various serialization techniques were tested, but the original structured format proved most effective. The initial causal structure is heuristically determined (e.g., Hill climbing (37)). The task instruction guides the goal, such as *"produce accurate and convincing synthetic data"*. Through adversarial optimization, the causal structure and instructions are refined to reduce the discriminator's accuracy. Thus, for each iteration $i$, $\theta_i$ is:

$$\theta_i = [\text{context}][\text{schema}], [\text{causal structure}]_i[\text{task instruction}]_i. \tag{1}$$

Note that subscription for iteration $i$ will be omitted for simplicity without loss of generalizability. Also, note that we used causal structure as a means to convey the relationship between variables within the prompt; thus, obtaining ground-truth causal structure is not our ultimate goal.

**Few shot examples.** The data generation process $\theta$ is supplemented with $n$ examples to leverage in-context few-shot learning. Structured data $(\mathbf{x}, y)$ is serialized into JSON format, e.g., *{age: 53, work class: self-emp, ...}* (Supplement listing 2 Lines 25-28). Various natural language serializations were tested but had minimal impact on performance. The number $n$ of examples is crucial; a large $n$ allows learning from diverse examples but is constrained by context length, while a small $n$ avoids overflow but underutilizes data. Our solution, "batches in a batch," splits a batch into smaller pieces that fit the input token size, generates a set of synthetic data, and collates them into $D_{\text{syn}}$ (see Algorithm 1 Line 6). This approach balances the trade-offs in in-context few-shot learning. The final input to the generator LLM is:

$$\theta_i, [JSON((\mathbf{x}, y)_1)], [JSON((\mathbf{x}, y)_2)], ..., [JSON((\mathbf{x}, y)_n)] \tag{2}$$

for each optimization iteration. See Supplement listing 2 for a full example.

**LLM as generator.** With the prompt in Eq. 2, the pre-trained, frozen LLM (e.g., GPT-3.5) generates synthetic data. The goal is to create similar but not identical text to the $n$ samples, with the temperature parameter controlling variability. The temperature is set low enough to maintain the original data distribution but high enough to avoid copying. The generator LLM runs multiple times with smaller examples in a batch, and the generated data is collated into $D_{\text{syn}}$. $D_{syn,i}$ denotes the synthetic data generated at iteration $i$. See Supplement listing 3 for an example.

**Conditional sampling.** As `MALLM-GAN` generates synthetic data using LLM, it seamlessly inherits the benefits of LLM, such as conditional sampling. LLM predicts the next tokens given a user-provided context, even when the specified condition is rare or given as a range. To conditionally sample the synthetic data, we modify the task instruction to contain specific conditions (Supplement Listing 4).

#### 3.2.2 DISCRIMINATOR

Based on the generated data, we evaluate and score the quality of $\theta$ by assessing how easy it is to distinguish generated synthetic data from real data. Naturally, this is a supervised learning rather than a reasoning task with LLMs. We build a discriminator $f$ such that $f : \mathcal{X} \to c$ where $\mathbf{x} \in \mathcal{X}$ and $f(\mathbf{x})$

is the predicted label $c$, which is 1 if $\mathbf{x} \in D_{train}$ and 0 if $\mathbf{x} \in D_{syn}$. Specifically, at each iteration $i$, a new set of synthetic data $D_{syn,i}$ is generated. We form the combined dataset $D_{train} \cup D_{syn,i}$. We assign labels to the combined dataset by

$$D_i = \{(\mathbf{x}, c) \mid \mathbf{x} \in D_{train}, c = 1\} \cup \{(\mathbf{x}, c) \mid x \in D_{syn,i}, c = 0\}.$$

We update the discriminator $f_i$ incrementally based on $f_{i-1}$. We evaluate the accuracy of the discriminator with $D_{test}$ and pass a pair of $(\theta_i, L(f_i))$ to the optimizer where $L(f)$ denotes the discriminatory power of $f$ (e.g., accuracy, likelihood). We prefer to use accuracy (rather than likelihood) because this is a direct measurement we aim to increase and because our optimizer does not require numerical derivatives.

The discriminator obtains better discriminatory accuracy to distinguish real or synthetic data as the discriminator accumulates the discriminatory power of past iterations $0, ..., i-1$ and is updated with newly generated, more realistic synthetic data from the current iteration $i$. However, on the other hand, as the $D_{syn}$ becomes more realistic over the iterations, it gets easier to fool the discriminator, and the discriminator's accuracy decreases. Therefore, our discriminator obtains better discriminatory power during this adversarial optimization.

### 3.2.3 OPTIMIZER

The next task is to optimize $\theta_i$ based on its score $L(f_i)$. Our parameter to optimize is $\theta$, a text, which doesn't have derivatives. So we use Optimization by Prompting, which leverages LLM as an optimizer (43). To make LLM acts as a optimizer, we provide a meta-prompt, which consists of the causal structure and the optimization task descriptions such as *"Your task is to optimize prompts for generating high-quality synthetic data. Aim ... "* (see Example in Supplement listing 5 Line 3-6).

To leverage LLM's in-context few-shot learning in the optimizer (43), we provide a few "examples" of possible solutions $(\theta, L(f))$. Note that the example here is different from data $(\mathbf{x}, y)$. We keep the top $k$ solution pairs over the past iteration as the optimization trajectory to guide the optimization. We sort the score, so that the more desirable $\theta$ goes to the end of prompt. This will allow the LLM to recognize patterns among the data generation process with better score. See example in Supplement listing 5 Line 9-31.

A potential pitfall is that the $L(f)$ of past iteration $0, ..., i-1$ is not comparable to the $L(f)$ of current iteration $i$. The past discriminators $f_0, ...f_{i-1}$ have much lower performance in discriminating real and fake, thus the score $L(f_0), ...L(f_{i-1})$ are not reliable to compare $\theta$ of past iterations with $\theta$ of current iteration. Thus we adjust the score $L(f)$ of past iterations with the latest discriminator $f_i$, so that all the scores are directly comparable to select best $\theta$.

In all, the optimizer LLM takes as input the meta prompt and a series of data generation process $\theta$ and adjusted scores $L(f_i)$). The optimizer outputs the revised data generation process, particularly focusing on causal structure and task instruction. We repeat the iterative optimization and generation until reaching to the maximum iteration.

```python
def optimize_theta(theta):
    theta_score_pairs = []
    for _ in range(max_epoch):
        for batch in D_train:
            # 1. Run generator
            D_syn = [LLM_generator(theta + example) for example in
                batch]
            # 2. Run discriminator
            labels_syn, labels_train = [0] * len(D_syn), [1] * len(
                D_train)
            (train, test), (train_label, test_label) =
                train_test_split(concat(D_train, D_syn), concat(
                labels_train, labels_syn))
            discriminator.update(train, train_label)
            score = get_accuracy(discriminator.predict(test),
                test_label)
            # 3. Run optimizer
            theta_score_pairs.append((theta, score))
            theta = LLM_optimizer(instruction + str(theta_score_pairs
                ))
```

```
15      return theta
16
17  def generate_synthetic_data(theta):
18      return [LLM_generator(theta + example) for example in D_train]
```

Listing 1: Python style pseudocode for `MALLM-GAN`'s optimization and generation

### 3.3 COMPARISON TO GAN AND CONVERGENCE

`MALLM-GAN`'s adversarial training is motivated by GAN, but it differs fundamentally from traditional GANs in that it operates in a natural language optimization space, using LLMs for prompt-based generation, which lacks formal mathematical guarantees. Unlike gradient-based optimization in GANs, `MALLM-GAN`'s optimization relies on empirical refinement of prompts through adversarial optimization with a discriminator. Theoretical convergence analysis is challenging due to the absence of numerical gradients. However, empirical convergence, demonstrated in our experiments in Section 4 and prior work (43), shows stable convergence, where discriminator accuracy declines as prompts are refined. This practical convergence criterion serves as a reliable alternative to formal guarantees in real-world tasks.

## 4 EXPERIMENTS

We present the evaluation results of `MALLM-GAN`. Our extensive experiments demonstrated that `MALLM-GAN` outperforms baselines in generating high-quality synthetic data while preserving data privacy, thanks to the adversarial optimization of the data generation process. Additionally, `MALLM-GAN`' provides explainable data generation through natural textual representation, effectively generating high-quality synthetic data based on user-provided conditions, even for rare categorical values or numeric ranges.

### 4.1 SETTING

**LLM.** We used HIPPA-compliant Azure OpenAI GPT-3.5(7) as our generator and GPT-4 (25)(gpt-4-32k-0613) as our optimizer. Due to the extensive workload of the generator, we opted for the lighter, faster gpt-35-turbo-0125 model with a 16k context length. For the optimizer, requiring combinatorial search and high-level reasoning, we used the up-to-date gpt-4-32k-0613 model. As the optimizer requires more "creativity" than the generator, the generator's temperature was set to 0.5, and the optimizer's to 1.0 after multiple trials.

**Discriminator.** Strong discriminators do not always contribute to a better generator (3). We tested Logistic regression, XGBoost, and neural network; we used the logistic regression model because it showed the highest performance while ensuring tractability during incremental updates over the iterations (Supplementary 6).

**Data.** Our benchmarks include several datasets from various domains: three public datasets (Adult(4), Medical Insurance(1), Asia(30)), and two private medical datasets (ATACH2, ERICH) (22). To ensure fair comparison without memorization concerns of LLM (e.g., public datasets are in the training corpus of LLM), private datasets were included. Details are in Supplement Table 5.

**Baselines.** We compared `MALLM-GAN` with multiple state-of-the-art tabular generative models such as: traditional over-sampling techniques, SMOTE (9), the variational auto-encoder, TVAE (42), the generative adversarial network, CTGAN (42), LLM-based synthetic data generation model, Be-GReaT(6), and a diffusion model, TabDDPM (19). Similar to `MALLM-GAN`, a prior work (31) uses in-context few-shot learning of pre-trained LLMs but incorporates post-hoc data selection, which is beyond our scope. A comparison without post-hoc selection is available in Table 3.

**Other hyperparameters.** Various serialization techniques have been proposed to transform tabular data into natural language text (14). We tested several serializations, such as Manual Template (14) (*"Age is 53, Work class type is self-employed, ..."*), which proved ineffective for moderate feature sizes; this serialization made the input prompt lengthy and talkative, only worked when feature size $|\mathbf{x}|$ is very small. Feature order permutation (6) also had negligible impact on performance. Specific hyperparameters and computing resources are available in Supplement Section 2.

**Training data size vs. quality of synthetic data**. We evaluated the impact of training data size $N = |D_{\text{train}}|$ on synthetic data quality by sampling subsets of different sizes ($N = 100, 200, 400, 800$).

We particularly aimed to compare performances in low and moderate data size. For fair comparison between real and synthetic data, synthetic data was generated to match the size of real data ($|D_{train}| = |D_{syn}|$). We held out 200 samples as the test set $D_{test}$ before sampling and replicated experiments for each sub sample five times to estimate the standard error of the evaluation metrics. The batch size was set to be 50, with maximum iterations set to be 5, 4, 3, 2 for data sizes $N = 100, 200, 400, 800$, respectively.

## 4.2 PERFORMANCE EVALUATION

We evaluate the performance of synthetic data generation models from two perspectives: Privacy leakage by Distance to Closest Records (DCR) and Machine Learning Efficiency (MLE) (10; 42).

**MLE.** To assess the utility of our synthetic data, we use it to train supervised model and test prediction accuracy on real data ($D_{test}$). The Adult data is used for a classification task, while the other three datasets are used for regression. For classification, we fit logistic regression model, random forest, and Support Vector Machine model, XGBoost Classifier, calculating F1 score. For regression, we fit linear regression, random forest, XGBoost Regressor and calculate $R^2$. For each model, we report the average of the best scores for each random seed. We also fit models using real data $D_{train}$ as a gold standard of the MLE for comparison. As a result, MALLM-GAN generated high-quality synthetic tabular data across multiple datasets and training data size, outperforming baselines (Table 2), specially with small training sizes ($N = 100$). This indicates MALLM-GAN's robustness to smaller sample sizes, unlike baselines that require more data. MALLM-GAN also outperformed baselines on both public and private datasets, suggesting it does not rely on the pre-trained LLM's memorization.

Table 2: Benchmark MLE results over 5 datasets. Baseline results were obtained from training the supervised models directly on the real data. SMOTE* interpolates data within the training set, thus it gets higher accuracy by copying training data and compromising DCR.

| | | Public dataset | | | Private dataset | |
| --- | --- | --- | --- | --- | --- | --- |
| | | Adult ($F1$) | Asia ($F1$) | Insurance($R^2$) | ATACH($R^2$) | ERICH($R^2$) |
| N=100 | Real data | 0.86 | 0.83 | 0.82 | 0.26 | -0.04 |
| | SMOTE* | $0.78 \pm 0.01$ | $0.83 \pm 0.00$ | $0.80 \pm 0.01$ | $0.27 \pm 0.03$ | $-0.15 \pm 0.13$ |
| | TabDDPM | $0.75 \pm 0.01$ | - | $-5.26 \pm 0.42$ | $-0.99 \pm 0.33$ | $-0.19 \pm 0.05$ |
| | CTGAN | $0.66 \pm 0.06$ | $0.63 \pm 0.19$ | $-0.09 \pm 0.11$ | $-0.40 \pm 0.21$ | $-0.33 \pm 0.11$ |
| | TVAE | $0.67 \pm 0.05$ | $0.83 \pm 0.01$ | $0.39 \pm 0.15$ | $-0.01 \pm 0.07$ | $-0.11 \pm 0.12$ |
| | Be-GReaT | $0.71 \pm 0.03$ | $0.83 \pm 0.00$ | $0.54 \pm 0.10$ | $-0.25 \pm 0.23$ | $-0.38 \pm 0.12$ |
| | MALLM-GAN | $\mathbf{0.79 \pm 0.02}$ | $\mathbf{0.83 \pm 0.00}$ | $\mathbf{0.72 \pm 0.00}$ | $\mathbf{0.27 \pm 0.07}$ | $\mathbf{-0.03 \pm 0.07}$ |
| N=200 | Real data | 0.85 | 0.83 | 0.83 | 0.27 | 0.16 |
| | SMOTE* | $0.78 \pm 0.04$ | $0.83 \pm 0.00$ | $0.79 \pm 0.02$ | $0.31 \pm 0.04$ | $0.05 \pm 0.06$ |
| | TabDDPM | $0.60 \pm 0.15$ | - | $0.56 \pm 0.14$ | $-0.55 \pm 0.33$ | $-0.30 \pm 0.06$ |
| | CTGAN | $0.61 \pm 0.02$ | $0.71 \pm 0.10$ | $-0.12 \pm 0.08$ | $-0.27 \pm 0.05$ | $-0.19 \pm 0.10$ |
| | TVAE | $0.67 \pm 0.05$ | $0.82 \pm 0.01$ | $0.62 \pm 0.05$ | $0.08 \pm 0.06$ | $-0.08 \pm 0.07$ |
| | BeGReaT | $0.69 \pm 0.05$ | $0.82 \pm 0.00$ | $\mathbf{0.72 \pm 0.03}$ | $0.16 \pm 0.06$ | $-0.18 \pm 0.16$ |
| | MALLM-GAN | $\mathbf{0.77 \pm 0.03}$ | $\mathbf{0.83 \pm 0.01}$ | $0.69 \pm 0.04$ | $\mathbf{0.28 \pm 0.07}$ | $\mathbf{0.02 \pm 0.02}$ |
| N=400 | Real data | 0.83 | 0.84 | 0.85 | 0.31 | 0.18 |
| | SMOTE* | $0.85 \pm 0.03$ | $0.84 \pm 0.00$ | $0.83 \pm 0.00$ | $0.32 \pm 0.02$ | $0.07 \pm 0.05$ |
| | TabDDPM | $\mathbf{0.82 \pm 0.03}$ | - | $\mathbf{0.79 \pm 0.03}$ | $0.36 \pm 0.02$ | $\mathbf{0.09 \pm 0.04}$ |
| | CTGAN | $0.63 \pm 0.02$ | $0.59 \pm 0.17$ | $-0.18 \pm 0.10$ | $-0.08 \pm 0.07$ | $-0.24 \pm 0.10$ |
| | TVAE | $0.71 \pm 0.07$ | $0.71 \pm 0.07$ | $0.62 \pm 0.05$ | $0.16 \pm 0.08$ | $-0.19 \pm 0.06$ |
| | Be-GReaT | $0.79 \pm 0.04$ | $0.79 \pm 0.00$ | $0.72 \pm 0.03$ | $0.20 \pm 0.06$ | $-0.13 \pm 0.07$ |
| | MALLM-GAN | $0.79 \pm 0.02$ | $\mathbf{0.83 \pm 0.00}$ | $0.71 \pm 0.03$ | $\mathbf{0.27 \pm 0.04}$ | $0.02 \pm 0.03$ |
| N=800 | Real data | 0.71 | 0.84 | 0.85 | 0.40 | 0.21 |
| | SMOTE* | $0.71 \pm 0.03$ | $0.84 \pm 0.00$ | $0.83 \pm 0.00$ | $0.37 \pm 0.03$ | $0.10 \pm 0.05$ |
| | TabDDPM | $0.70 \pm 0.03$ | - | $\mathbf{0.83 \pm 0.01}$ | $-0.53 \pm 0.45$ | $\mathbf{0.12 \pm 0.04}$ |
| | CTGAN | $0.64 \pm 0.05$ | $0.48 \pm 0.06$ | $-0.41 \pm 0.06$ | $-0.05 \pm 0.06$ | $-0.04 \pm 0.02$ |
| | TVAE | $0.77 \pm 0.02$ | $0.82 \pm 0.01$ | $0.68 \pm 0.01$ | $0.12 \pm 0.07$ | $-0.05 \pm 0.03$ |
| | Be-GReaT | $0.75 \pm 0.07$ | $0.82 \pm 0.00$ | $0.53 \pm 0.21$ | $0.00 \pm 0.07$ | $-0.04 \pm 0.05$ |
| | MALLM-GAN | $\mathbf{0.80 \pm 0.02}$ | $\mathbf{0.84 \pm 0.00}$ | $0.72 \pm 0.01$ | $\mathbf{0.36 \pm 0.02}$ | $0.02 \pm 0.02$ |

**DCR distributions.** The DCR metric assesses the realism and diversity of synthetic data. It determines whether synthetic data points are too similar to the real data points (potential privacy leakage) or too dissimilar (hurting the utility of the synthetic data). The DCR is defined as $d(\mathbf{x}_{syn}, D_{real}) = \min_{\mathbf{x}_{real} \in D_{real}} l_1\text{-norm}(\mathbf{x}_{syn}, \mathbf{x}_{real})$. Low DCR indicates that synthetic data are very close to real data points, implying a privacy leakage, as synthetic data too closely mimic the real data. We chose to use $l_1$-norm distance to measure the distance between two data points (6). For the categorical variables, the distance is 1 if two categories are different; otherwise, 0. As a result,

`MALLM-GAN` achieved similar or higher DCR levels compared to baseline models (Fig. 2), implying effective privacy protection without compromising MLE. Overall, `MALLM-GAN` demonstrated superior performance in generating synthetic data with small data as balancing privacy and utility.

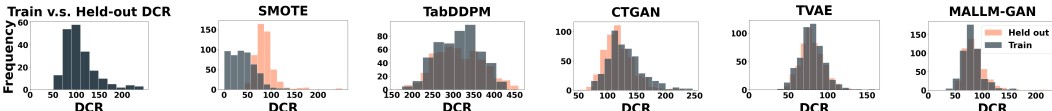

Figure 2: DCR between the synthetic data and the real data. DCR were calculated based on training data and held-out test data for each model. A good model should have similar distributions between the DCR to training and the DCR to held-out dataset.

### 4.3 ABLATION STUDY

**Number $n$ of example in in-context few shot learning**. Due to the LLM's limited context length, we implemented a"batches in a batch" method to leverage all training data within these constraints (Section 3.2.1). We varied the number $n$ of examples and found $n = 1$ to be optimal, achieving high DCR without compromising MLE (Supplement Section 2.5).

**Causal structure and Optimization**. To assess the impact of each component on overall performance, we examined the contribution of the causal structure in the data generation process $\theta$ and the LLM as an optimizer. We compared the full model, which includes both components, to a version without them, similar to CLLM (31) without post-processing data selection (Table 3). The ablation study showed that incorporating the causal structure alone did not significantly improve the MLE compared to a model with only in-context few-shot learning. However, the LLM optimizer improved $\theta$ using prior knowledge encoded in LLM and finally achieved the highest MLE. Incorporating external knowledge into LLMs has been shown to significantly improve the quality of generated text, similar to retrieval-augmented generation (RAG) (20). Our approach shares this concept by incorporating a "knowledge" graph but optimizes the knowledge itself through adversarial optimization.

Table 3: MLE of ablated models to evaluate the effects of causal structure in data generation process and optimization via LLM. Causal: Causal structure in data generation process, Opt: Optimization by LLM.

|  | Few-shot | Few-shot+Causal | Few-shot+Causal+Opt (ours) |
|---|---|---|---|
| Adult ($F1$) | $0.7550 \pm 0.0454$ | $0.7503 \pm 0.0393$ | $\mathbf{0.7892 \pm 0.0358}$ |
| Asia ($F1$) | $0.2335 \pm 0.0000$ | $0.2756 \pm 0.2842$ | $\mathbf{0.8282 \pm 0.0041}$ |
| Insurance ($R^2$) | $0.6821 \pm 0.0193$ | $0.6718 \pm 0.0916$ | $\mathbf{0.7152 \pm 0.0447}$ |
| ATACH ($R^2$) | $0.1581 \pm 0.0850$ | $0.1326 \pm 0.0637$ | $\mathbf{0.2726 \pm 0.0707}$ |
| ERICH ($R^2$) | $-0.0647 \pm 0.0701$ | $\mathbf{0.0281 \pm 0.0424}$ | $-0.0253 \pm 0.0671$ |

### 4.4 OPTIMIZATION TRAJECTORY OF DATA GENERATION PROCESS

A key advantage of `MALLM-GAN` is its transparent data generation process, described in natural text, which allows us to observe the evolution trajectory of the data generation mechanism during adversarial optimization. We present examples of optimization trajectories. We showed how the causal structure evolves to ground truth (Fig. 3) over iteration. We used the Asia dataset because it has known ground-truth causal structures and reported graph edit distance (GED) between ground truth and identified causal structures. In this example, the heuristically initialized causal structure gradually converges to ground truth, thanks to the knowledge obtained from the pre-trained LLM. Different convergence patterns were observed with different initialization strategies (Table 9), supporting the benefit of our heuristic initialization. We also investigated how the task instruction in the generator prompt gets sophisticated and how the discriminator's accuracy changes over iterations. In Table 4, the task instructions evolved to include specific details, and the discriminator's accuracy decreased, implying that synthetic data gets indiscriminative to real data.

### 4.5 CONDITIONAL SAMPLING

We leverage the generator's conditional generative capability to create synthetic data with user-provided conditions, focusing on categorical values and numerical ranges. We compare `MALLM-GAN`

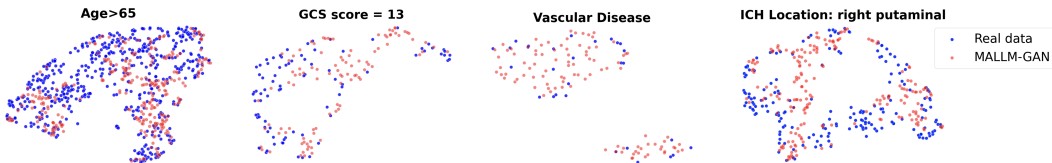

Figure 3: An example of trajectory of causal structure in data generation process over adversarial optimization using Asia dataset. **T**: Tuberculosis, **V**: Visit to Asia, **S**: Smoke, **LC**: Lung cancer, **T/L**: Tuberculosis or Lung cancer, **CX**: Chest X-ray, **D**: Dyspnea, **B**: Bronchitis

| Iteration | Task instruction | Score |
|---|---|---|
| Epoch 1 | *"The ultimate goal is to produce accurate and convincing synthetic data that dutifully represents these causal relationships given the user provided samples."* | 100.0% |
| Epoch 2 | *"The ultimate goal is to create a detailed and convincing dataset that accurately mirrors these causal pathways. While synthesizing your data, keep in mind the following key relationships: a 'visit to Asia' increases the likelihood of 'tuberculosis', 'smoking' can lead to 'lung cancer' and 'bronchitis', and both 'tuberculosis' and 'lung cancer' can contribute to 'either tuberculosis or lung cancer', which in turn can lead to 'Dyspnea'. Also, take note of how both 'tuberculosis' and 'lung cancer' are associated with 'chest X-ray' results. Your data should reflect these intricate relationships while remaining consistent and realistic."* | 76.19% |
| Epoch 4 | *"You are tasked with generating a synthetic dataset that faithfully demonstrates the given causal connections. Make sure the dataset illustrates how a 'visit to Asia' can cause 'tuberculosis', how 'smoking' can lead to 'lung cancer' and 'bronchitis', and how either 'tuberculosis' or 'lung cancer' can eventually incite 'Dyspnea'. Also, the dataset should reasonably reveal how a 'chest X-ray' ties in with 'tuberculosis' and 'lung cancer'. Ensure the synthetic data reflects realistic scenarios where these factors interact, affecting each other exactly as per these defined causal relationships."* | 66.67% |

Table 4: Trajectory of task instruction in data generation process over adversarial optimization. Lower score is the better.

and baseline models by visualizing lower-dimensional projections (UMAP). For categorical conditions, we selected three rare conditions in the ERICH dataset: i) *hematoma location = right putaminal*, ii) *GCS score = 13*, and iii) *prior history in vascular disease*. The conditions were met by 187, 83, and 29 patients, respectively. All three baselines failed to generate synthetic data due to insufficient training data. In contrast, `MALLM-GAN` successfully generated data with distributions similar to the real data (Fig. 4). For numeric range conditions, we selected *'age'* > 65 in the ERICH dataset, met by 534 patients. The baselines were unable to incorporate numeric range conditions by design. However, `MALLM-GAN` successfully generated data satisfying the condition (Fig. 4), demonstrating its ability to understand and flexibly apply conditions in natural text format.

Figure 4: Real and synthetic data distribution with three categorical conditions and one numerical range condition in ERICH data.

## 4.6 LIMITATIONS

The proposed framework has several shortcomings. Firstly, due to the limited context length of the LLM, our model struggles with high-dimensional datasets with too many categorical variables, which make the context information lengthy and reduce the success rate of data generation. Another limitation of synthetic data generation introduced by LLM is that the LLM struggles with random

number generation as pointed out in (15), which cast negative effects on our framework's potential when dealing with datasets of many continuous variables. Additionally, mimicking the traditional GAN framework, it suffers from a theoretical convergence guarantee. While our model performs well with small sample sizes, showing better results than other baselines, the improvement diminishes with larger datasets. Moreover, the training and generation process is costly when dealing with large data volumes.

## 5 CONCLUSION

We propose a novel framework to generate synthetic tabular data by leveraging multi-agent LLMs to address the limited sample size issues that are prevalent in healthcare. Compared with other LLM-based methods, we propose an in-context learning approach that does not require fine-tuning on LLM but still leverages the whole data. We use causal structure to guide the data generation process and mimic a GAN architecture to optimize the process. We demonstrate that our model can generate high-quality synthetic data while preserving the privacy of real data. Moreover, compared with other black box models, our proposed work enables transparent data generation that allows domain experts to control the process.

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

SUPPLEMENTARY

# 1 EXAMPLE OF PROMPTS AND OUTPUT

Here, we provided example of generator prompt and optimizer prompt. Note that generator prompt evolves over the iteration.

```
1  System role:
2  % Specify role and task
3  You are a data generation model. Your task is to understand the
       instruction below and generate tabular data.
4
5  % Context of data
6  <context>The dataset include subject's social economic factors and
        demographics with the label that indicates whether their
       income is higher than 50k. </context>
7
8  % Data schema
9  <schema> age (numerical),workclass (categorical), education (
       categorical), education-num (numerical),marital-status (
       categorical), occupation (categorical), relationship (
       categorical), race (categorical), sex (categorical), capital-
       gain (numerical),capital-loss (numerical),hours-per-week (
       numerical),native-country (categorical), Income (categorical)
       </schema>
10
11 %Categorical variables and their available categories
12 <categorical variables> workclass: {'Private', 'Local-gov', '
       Without-pay', 'Self-emp-not-inc', 'State-gov', 'Federal-gov',
       'Self-emp-inc'}, education: {'Some-college', 'Masters', '11th',
        '1st-4th', '7th-8th', 'Bachelors', 'Doctorate', '12th', '5th
       -6th', 'Prof-school', 'Assoc-voc', 'Assoc-acdm', '10th', '9th',
        'HS-grad'}, marital-status: {'Divorced', 'Married-spouse-
       absent', 'Married-civ-spouse', 'Never-married', 'Widowed', '
       Separated'}, occupation: {'Handlers-cleaners', 'Transport-
       moving', 'Sales', 'Prof-specialty', 'Farming-fishing', '
       Machine-op-inspct', 'Adm-clerical', 'Other-service', 'Craft-
       repair', 'Protective-serv', 'Exec-managerial', 'Tech-support',
        'Priv-house-serv'}, relationship: {'Wife', 'Not-in-family', '
       Other-relative', 'Unmarried', 'Own-child', 'Husband'}, race: {
       'Black', 'Amer-Indian-Eskimo', 'Other', 'Asian-Pac-Islander',
       'White'}, sex: {'Male', 'Female'}, native-country: {'Vietnam',
        'Mexico', 'Hong', 'Taiwan', 'Italy', 'Portugal', 'Ireland', '
       Guatemala', 'El-Salvador', 'United-States'}, Income: {'>50K',
       '<=50K'}
13 </categorical variables>
14
15 %causal structure
16 <causal structure> Consider this optimized causal graph of the
       data, where a pair (A, B) is used to represent a scenario
       where A affects B: [('age', 'workclass'), ('education', '
       education-num'), ('education-num', 'Income'), ('marital-status
       ', 'relationship'), ('occupation', 'Income'), ('hours-per-week
       ', 'Income'), ('workclass', 'Income')]
17
18 This adjusted graph introduces 'education-num', which is a key
       determinant of 'Income'. Be sure to reflect 'age' impact on '
       workclass' and 'marital-status' effect on 'relationship'. When
        creating the 'Income' data, pay careful attention to the
       roles of 'education', 'education-num', 'occupation', and '
       hours-per-week' as stated in the causal graph.
19 </causal structure>
20
21 %Task
```

```
22  <task> The ultimate goal is to produce accurate and convincing
        synthetic data that dutifully represents these causal
        relationships. As such, strive for a quality score that is
        less than 70.0%. </task>
23
24  User role:
25  % Example
26  <example> Here are examples from real data:
27  [{'age': 53.0, 'workclass': 'Self-emp-not-inc', 'education': '10th
        ', 'education-num': 6.0, 'marital-status': 'Married-civ-spouse
        ', 'occupation': 'Farming-fishing', 'relationship': 'Husband',
         'race': 'White', 'sex': 'Male', 'capital-gain': 0.0, 'capital-
        loss': 0.0, 'hours-per-week': 60.0, 'native-country': 'United-
        States', 'Income': '<=50K'}, {'age': 23.0, 'workclass': '
        Private', 'education': 'HS-grad', 'education-num': 9.0, '
        marital-status': 'Never-married', 'occupation': 'Adm-clerical',
         'relationship': 'Own-child', 'race': 'White', 'sex': 'Female',
         'capital-gain': 0.0, 'capital-loss': 0.0, 'hours-per-week':
        40.0, 'native-country': 'United-States', 'Income': '<=50K'}]
28  </example>
29
30  <instruction>
31  Generate two synthetic samples mimic the provided samples. DO NOT
        COPY the samples and try to make the generated samples diverse.
         The response should be formatted strictly as a list in JSON
        format, suitable for direct use in data processing scripts
        such as conversion to a DataFrame in Python. No additional
        text or numbers should precede the JSON data.
32  </instruction>
```

Listing 2: Example of generator prompt

```
1  json
2  [{"treatment": 0, "age": 68.2, "ICH volume": 4.1, "ICH Location":
        "L Lobar", "IVH volume": 0.2, "GCS score": 14.3, "NIHSS score":
         11.7, "Systolic blood pressure": 195.0, "Diastolic Blood
        Pressure": 83.0, "Hypertension": 1, "Hyperlipidemia": 1, "Type
         I Diabetes": 0, "Type II Diabetes": 0, "Congestive heart
        failure": 0, "Atrial Fibrillation": 0, "PTCA": 0, "Peripheral
        Vascular Disease": 0, "Myocardial fraction": 0, "Anti-diabetic
        ": 0, "Antihypertensives": 1, "White blood count": 4.3, "
        Hemoglobin": 12.5, "Hematocrit": 37.7, "Platelet count": 129.0,
         "APTT": 35.3, "INR": 1.1, "Glucose": 148.0, "Sodium": 145.0,
        "Potassium": 4.1, "Chloride": 106.0, "CD": 30.1, "Blood urea
        nitrogen": 18.0, "Creatinine": 1.2, "race": "White", "sex": "
        Female", "ethnicity": "Hispanic", "mRS score after 30 days":
        2.7}]
```

Listing 3: Example of generator output. We presented an example with sufficiently high DCR (39.7) to protect patient data privacy

```
1  <Instruction>
2  Generate {number of samples in real data meeting the conditions}
        synthetic samples with {user-provided conditions}. Response
        should be formatted strictly as a list in JSON format,
        suitable for direct use in data processing scripts such as
        conversion to a DataFrame in Python. No additional text or
        numbers should precede the JSON data.
3  </Instruction>
```

Listing 4: Modified instruction in generator prompt for conditional sampling

```
1  System role:
2  % Specify role and task
```

```
3   Your task is to optimize prompts for generating high-quality
        synthetic data. Aim to lower the scores associated with each
        casual structure and prompt, where a lower score reflects
        better quality. Here are the steps:
4   1. Examine the existing prompt-score pairs.
5   2. Adjust the causal graph to better represent the underlying
        relationships by adding or removing connections, and consider
        incorporating new features from the list {self.cols}.
6   3. Modify the prompt guidance to align with the revised causal
        graph, ensuring it aids in reducing the score.
7
8   User role:
9   <pair>
10  Reflecting the adjusted causal graph of the data, where each tuple
        (A, B) indicates that A impacts B:
11  [('age', 'workclass'), ('marital-status', 'relationship'), ('
        marital-status', 'Income'), ('relationship', 'sex'), ('
        education', 'Income'), ('occupation', 'Income'), ('workclass',
        'Income'), ('hours-per-week', 'Income')]
12
13  Use this causal graph as a guide to generate synthetic data that
        closely mirrors the real-world dataset. Remember to factor in
        the influence of 'age' on 'workclass', and 'marital-status' on
        'relationship' and 'Income'. The 'relationship' should guide
        the generation of the 'sex' attribute. Further, take into
        consideration the effects of 'education', 'occupation', and '
        hours-per-week' on 'Income' when synthesizing your data. The
        goal is to produce synthetic data that convincingly mimic
        these causal relationships.
14  Set your aim to achieve a score below 75.0%.
15  Score: 80.0%
16  </pair>
17
18  <pair>
19  Consider the revised and detailed causal graph of the data, which
        includes ('age', 'workclass'), ('marital-status', '
        relationship'), ('relationship', 'sex'), ('education', 'Income
        '), ('occupation', 'Income'), ('workclass', 'Income'), ('hours-
        per-week', 'Income'):
20
21  In light of the causal graph, generate synthetic samples that
        mimic the structure in the provided dataset. Values such as '
        age' should reflect on 'workclass'; 'marital-status' and '
        relationship' should collaborate to inform 'sex', while '
        education', 'occupation', 'workclass', and 'hours-per-week'
        should exhibit their influence on 'Income'. Also consider '
        marital-status' influence on 'Income'. Your aim is to generate
        synthetic data that fully embody the interconnections within
        this causal graph.
22  Aim to achieve a score lower than 75%
23  Score: 80.95%
24  </pair>
25
26  <pair>
27  Here is the causal graph of the data, where a tuple (A, B)
        indicates A causes B:
28  [('marital-status', 'relationship'), ('marital-status', 'Income'),
        ('relationship', 'sex')]
29  Given the description of the data, generate synthetic samples that
        mimic the provided samples.
30  Score: 85.71%
31  </pair>
32
```

```
33   Your updated prompt should explicitly include any modifications to
        the causal graph and guidance. The aim is to create a prompt
        that leads to the lowest possible score.
34
35   The updated prompt:
```

Listing 5: Example of optimizer prompt

```
1   <Causal structure> The optimized causal network, suggesting the
        influence of variable A on variable B, includes the following
        relationships: [('Age', 'Hyperlipidemia'), ('Hyperlipidemia',
        'Type II Diabetes'), ('Type II Diabetes', 'Blood urea nitrogen
        '), ('Blood urea nitrogen', 'Creatinine'), ('Hypertension', '
        Congestive heart failure'), ('Congestive heart failure', '
        Atrial Fibrillation'), ('Atrial Fibrillation', 'GCS score'), (
        'GCS score', 'mRS score after 30 days'), ('Anti-diabetic', '
        Type I Diabetes'), ('Type I Diabetes', 'Antihypertensives'), (
        'Antihypertensives', 'Potassium'), ('Potassium', 'Sodium'), ('
        PTCA', 'Peripheral Vascular Disease'), ('Peripheral Vascular
        Disease', 'Myocardial fraction'), ('Myocardial fraction', '
        Hemoglobin'), ('Hemoglobin', 'Hematocrit'), ('race', '
        ethnicity'), ('Sex', 'Hyperlipidemia')]</Causal structure>
2
3   <Task> Your task is to create realistic synthetic patient data,
        keeping the altered causal relationships as your guiding
        principle. Ensure the data reflects a diverse set of potential
         patient scenarios, evidencing the variety of health
        conditions one might find in a clinical setting. Remember that
         the engineered data should present unique, individual patient
         scenarios, each portraying a different, complex clinical
        situation. The synthetic data needs to be representative of
        different demographics ('Sex', 'race', 'ethnicity') and should
         also take into consideration different health conditions and
        treatment plans.</Task>"
```

Listing 6: Example of optimizer output

## 2 EXPERIMENT DETAILS

### 2.1 BENCHMARK DATASETS DESCRIPTIONS

We provide detailed description on the benchmark data in Table 5

Table 5: Datasets description

| | # samples | # features | Description | Source |
|---|---|---|---|---|
| Adult | 32,561 | 14 | The dataset include people's social economic factors and demographics with the label that indicates whether their income is higher than 50k. | (4) |
| Medical Insurance | 2,772 | 7 | This is a dataset used to describe the paitents' demographics with their health insurance bills. | (1) |
| Asia | 10000 | 8 | This is the dataset used to illustrate the utility of Baysian network to do causal structure discovery. The dataset is available in the R-package(30). | |
| ATACH2 | 1,000 | 37 | This is an RCT data that investigate in treatment for Intracerebral hemorrhage patients. | (27) |
| ERICH | 1,521 | 29 | The data is from a case-control study of Intracerebral Hemorrhage study which aims to investigate in the Ethnic/Racial variations. | (39) |

## 2.2 HYPERPARAMETERS

Specific hyperparameters for each model are provided below.

- **CTGAN**: Default parameters

- **TVAE**: Default parameters

- **BeGReaT**:

    - Base LLM: Distiled-GPT2

    - Batch size: 40

    - Epochs: Depend on the feature numbers and the total sample size. (200-400)

- **MALLM-GAN**:

    - Temperature for generator: 0.5

    - Temperature for optimizer: 1.0

    - Batch size: 50

    - Discriminator: XGBoost (max depth: 3, eta: 0.3, objective: binary:logistic)

- **TabDDPM**: Default parameters

## 2.3 COMPARISON AMONG DIFFERENT KINDS OF DISCRIMINATORS

Table 6: Comparison of different discriminators effects on the quality of the synthetic data. An experiments on sub-sample of Adult data.

|  |  | N = 100 | N = 200 | N = 400 | N = 800 |
|---|---|---|---|---|---|
|  | XGBoost | $0.78 \pm 0.03$ | $0.73 \pm 0.01$ | $0.76 \pm 0.06$ | $0.72 \pm 0.00$ |
| Adult (F1 score) | Logistic regression | $0.79 \pm 0.02$ | $\mathbf{0.77 \pm 0.02}$ | $\mathbf{0.79 \pm 0.03}$ | $\mathbf{0.80 \pm 0.02}$ |
|  | Neural Network | $\mathbf{0.80 \pm 0.02}$ | $0.57 \pm 0.12$ | $0.78 \pm 0.06$ | $0.67 \pm 0.12$ |

## 2.4 COMPUTING RESOURCE DETAILS

The model proposed in this study does not require extensive computing resource for fine-tuning. However, this model require access to Azure service. For other baseline models, they are implemented on one NVIDIA H100 80GB HBM3 GPU.

## 2.5 NUMBER OF EXAMPLES IN IN-CONTEXT FEW SHOT LEARNING

Given the limited context length that the LLM can understand, we proposed "batches in a batch" method to leverage all training data in limited context length in the generator LLM (Section 3.2.1). We varied the number $n$ of few-shot examples by $n = 1, ...5$ and measured the MLE (Fig. 5) and DCR distribution (Table 7, 8) to find the optimal number $n$. As a result, the increasing number $n$ of examples did not always increase the MLE of synthetic data (Fig. 5) but decreased the DCR (Table 7), thus increasing privacy concerns. Instead, $n = 1$ achieved sufficiently high DCR without compromising MLE. The MLE did not increase with more examples because the more examples will increase the context length and the generator LLM overlook some key context information. On the other hand, the DCR decreased with more examples because the generator LLM is more likely to stick to copy the provided examples. Interesting, the increasing number of examples does not affect the DCR of synthetic data generated from public dataset (Adult, Insurance).

## 2.6 OPTIMIZATION TRAJECTORY: EXAMPLE ON INSURANCE DATASET

As seen in Table 9, the causal structure does not converge to the ground truth after 5 epochs. When initialized with the ground truth, the causal structure maintains slight fluctuations. Another example of causal structure discovery on insurance dataset were presented in Supplementary 2.6.

Here is the example for the Insurance dataset. Initially, the causal structure derived from heuristics had no edges (Fig. 6). Over iterative optimization, a stable causal structure emerged. The task instructions evolved to include specific details (Table 10). Our objective was to reduce the discriminator's

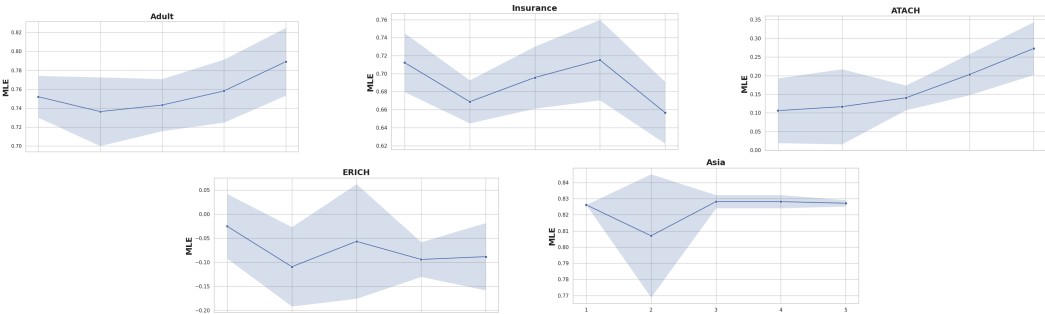

Figure 5: Number $n$ of examples and MLE.

|  | 1 | 2 | 3 | 4 | 5 |
|---|---|---|---|---|---|
| Adult | 5, 6, 10 | 5, 7, 12 | 4, 6, 9 | 4, 6, 10 | 4, 6, 11 |
| Insurance | 31, 93, 301 | 44, 66, 453 | 32,60, 182 | 33, 73, 167 | 29, 55, 168 |
| Asia | 0, 0, 0 | 0, 0, 0 | 0, 0, 0 | 0, 0, 0 | 0, 0, 0 |
| ATACH2 | 61, 73, 88 | 72, 87, 97 | 69, 78, 89 | 66, 75, 94 | 67, 83, 104 |
| ERICH | 53, 61, 79 | 59, 78, 96 | 59, 74, 101 | 56, 72, 96 | 50, 64, 88 |

Table 7: Number $n$ of examples and DCR to **training dataset**. 25%, 50% (Median), 75% quantile.

accuracy, and the score decreased over iterations, thanks to the optimized causal structure and task instruction.

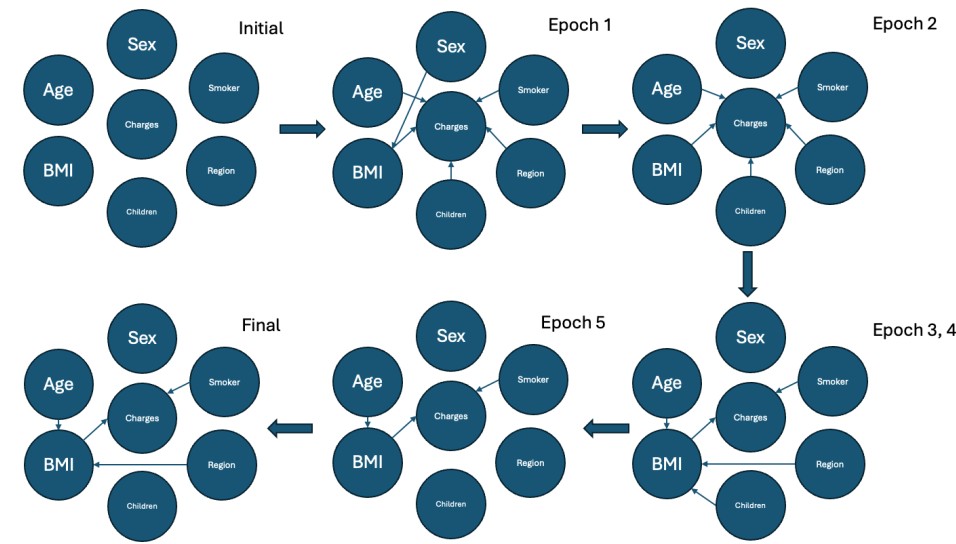

Figure 6: Trajectory of causal structure in data generation process over adversarial optimization.

Table 8: Number $n$ of examples and DCR to **held out dataset**. 25%, 50% (Median), 75% quantile.

|  | 1 | 2 | 3 | 4 | 5 |
|---|---|---|---|---|---|
| Adult | 4, 7, 10 | 5, 7, 11 | 5, 6, 10 | 4, 7, 10 | 4, 7, 11 |
| Insurance | 30, 115, 337 | 34, 91, 405 | 36, 76, 245 | 24, 64, 170 | 27, 70, 150 |
| Asia | 0, 0, 0 | 0, 0, 0 | 0, 0, 0 | 0, 0, 0 | 0, 0, 0 |
| ATACH2 | 84, 100, 120 | 82, 99, 122 | 81, 97, 125 | 79, 98, 124 | 82, 103, 128 |
| ERICH | 70, 87, 110 | 66, 82, 111 | 51, 82, 104 | 62, 80, 108 | 62, 80, 117 |

Table 9: Graph Edit Distance (GED) between causal structure in $\theta$ and ground truth

| Epoch | No Initialization | Heuristic Initialization | Ground Truth Initialization |
|---|---|---|---|
| 0 | 16 | 5 | 0 |
| 1 | 4 | 1 | 0 |
| 2 | 4 | 1 | 0 |
| 3 | 3 | 0 | 0 |
| 4 | 4 | 0 | 2 |

| Iteration | Task instruction | Score |
|---|---|---|
| Epoch 1 | *"The task is to generate synthetic data that accurately mirrors these causal relationships. The data should include variables such as 'age', 'sex', 'bmi', 'children', 'smoker', 'region', and 'charges'. Each variable should influence the others as per the causal structure, creating a realistic representation of possible real-world scenarios."* | 80.95% |
| Epoch 2 | *"Your task is to generate synthetic data that faithfully represents these causal relationships. The data should encompass variables such as 'age', 'bmi', 'children', 'smoker', 'region', and 'charges'. Each variable should affect the others in accordance with the causal structure, providing a credible simulation of potential real-world scenarios"* | 76.19% |
| Epoch 4 | *"The ultimate goal is to generate synthetic data that accurately reflects these causal relationships. The synthetic data should incorporate factors such as 'age', 'sex', 'bmi', 'children', 'smoker', 'region', and 'charges', and their influence on each other as indicated in the causal structure. The synthetic data should be convincing and representative of the real-world scenarios where these variables interact."* | 35.67% |

Table 10: Trajectory of task instruction in data generation process over adversarial optimization. Lower score is the better.

