# OpenReview forum: "MALLM-GAN: Multi-Agent Large Language Model as Generative Adversarial Network for Synthesizing Tabular Data"
_ICLR.cc/2025/Conference — Submitted to ICLR 2025_

### Official Review · Reviewer_fKif · 2024-11-02

**Soundness:** 3
**Presentation:** 3
**Contribution:** 3
**Rating:** 5
**Confidence:** 4

**Summary:**

The paper proposes a method for generating synthetic tabular data explicitly by leveraging the in-context learning of LLMs to mimic the adversarial process of GAN. This is achieved by prompting the LLM to build the generator, and then using a disriminator to identify the real data from synthetic data. The prompt of the generator is optimized by an LLM.

**Strengths:**

- Both the generation and optimization process are explicit, offering better explainability.

- The proposed pipeline enables automatic optimization and generation of synthetic data, which addresses the data scarcity problem of downstream tasks.

**Weaknesses:**

- The initial prompt for the generator and optimizer still requires empirical knowledge about the task and labor efforts, which makes application to eifferent tasks difficult.
- Although comparisons were made with multiple methods in the experiments, there is a lack of comparison with methods that use the same LLM (GPT-3.5) as data generator.

**Questions:**

The author(s) propose to intergrating the descriminator and optimization steps into the synthetic data curation process. My main concern is how this approach will improve the quality of synthetic data compared to  common LLM-based methods.
The author(s) should compare with other method using GPT-3.5 as the base model. For instance, the author(s) might consider comparisons against baselines such as human-written instructions, instructions generated directly by GPT-4, or examples randomly selected from the dataset as few-shot examples for synthetic data generation. These are essential for demonstrating the effectiveness of MALLM-GAN.
I would raise my score if the authors could provide more solid and fair comparative results to demonstrate the effectiveness of MALLM-GAN.

---

> ### Author Response · Authors · 2024-11-19
> **To reviewer fKif**
>
> We sincerely thank Reviewer hkAL for reviewing our paper and recognizing the novelty of our proposed method. Below, we address the two concerns you raised:
> - **Human effort for initial prompt.** We respectfully disagree with the reviewer in that the proposed model requires substantial effort (e.g., empirical knowledge) for initialization. Our model requires three inputs: 1) brief one-sentence data description, 2) variable name, and 3) initial causal structure (Eq (1)). Specifically, the causal structure is automatically initialized using a Bayesian Network algorithm, and other contextual information is automatically extracted from table column headings and cells. The only component requiring manual input is a brief one-sentence data description, which provides background context for the dataset. We don't see the one-sentence data description (which serves as a seed input for LLM to grow) as a substantial effort compared to the typical data-driven approach.
>
> - **Same LLM** Thank you for raising this concern. As noted in Lines 296–297,
> >   "We used HIPPA-compliant Azure OpenAI GPT-3.5(7) as our generator and GPT-4 (25)(gpt-4-32k-0613) as our optimizer,"
>
> we used GPT-3.5 as the data generator for our experiments, as the reviewer suggested. If the reviewer actually intended to suggest GPT-4 for the data generator, we cannot use GPT-4 for the data generator due to ethical concerns. We use privacy-sensitive patient data to avoid data leakage in evaluation. Only Azure OpenAI GPT-3.5 was a HIPAA-compliant LLM. It is our expectation that if we use GPT-4 for the data generator not GPT-3.5, our model's performance will only increase.
>
> **For the question**:
>
> This is a valid point, and the ablated comparison results that the reviewer suggested are already in Table 3. Let us clarify.
> - First column (Few-shot): the results with human-written instruction and manually selected few-shot examples (Line 398)
> - Second column (Few-shot+Causal): the results with the naive, heuristic causal structure without discriminator and optimizer.
> - Third column (Few-shot+Causal+Opt): Full results
>
> | Dataset (Metric) | Few-shot | Few-shot+Causal | Few-shot+Causal+Opt (ours) |
> |------------------|----------|-----------------|----------------------------|
> | **Adult (F1)**   | 0.7550 ± 0.0454 | 0.7503 ± 0.0393 | **0.7892 ± 0.0358**      |
> | **Asia (F1)**    | 0.2335 ± 0.0000 | 0.2756 ± 0.2842 | **0.8282 ± 0.0041**      |
> | **Insurance (R²)** | 0.6821 ± 0.0193 | 0.6718 ± 0.0916 | **0.7152 ± 0.0447**      |
> | **ATACH (R²)**   | 0.1581 ± 0.0850 | 0.1326 ± 0.0637 | **0.2726 ± 0.0707**      |
> | **ERICH (R²)**   | -0.0647 ± 0.0701 | **0.0281 ± 0.0424** | -0.0253 ± 0.0671    |
>
> The results demonstrate that when using GPT-3.5 solely as the generator, the quality of the synthetic data, as evaluated through downstream task performance, is notably poorer than the data generated with causal structure guidance and optimization in our model. These findings highlight the added value of our proposed method in enhancing data quality.
>
> We appreciate the reviewer for highlighting this point and welcome any further questions or suggestions for clarification.

---

### Official Review · Reviewer_hkAL · 2024-11-02

**Soundness:** 2
**Presentation:** 2
**Contribution:** 2
**Rating:** 5
**Confidence:** 4

**Summary:**

The paper proposes a novel GAN-inspired framework that leverages large language models (LLMs) as the generator and a classifier as the discriminator. Instead of optimizing at the model's weight level, the optimization occurs at the text prompt level, guiding the generation of synthetic data. The prompt used in data generation incorporates a natural language description that outlines the data collection context, the schema, relationships between columns (causal structure), and task instructions. Throughout training, the context and schema remain fixed, while the causal relationships and task instructions are refined to minimize the discriminator's accuracy.

The generation process begins with a few-shot setup to illustrate data structure and is followed by training the discriminator on both original and synthetic data. The discriminator is then evaluated on a held-out test set, and its performance score is provided to GPT-4 for further prompt optimization in the generator. This iterative feedback loop continues, where a series of top-performing discriminator scores are used by GPT-4 to refine the generator prompt, thereby enhancing synthetic data quality.

**Strengths:**

1. The experimental set up is quite innovative where a signal is sent to the generator regarding its generation quality via the llm optimizer which refines the prompt. This could potentially save a lot of compute as it makes quality generation possible without finetuning.
2. Using LLM’s to rewrite prompts based on signals in the form of scores is good.
3. Conditional generation through natural language appended to the prompt makes generation of simulations possible.

**Weaknesses:**

1. The claim made is that few-shot learning is not scalable due to the limited context length of the models in a data-scarce scenario and thus all the examples cannot be utilized by the model for generating new data. However models like Gemini are now available with context windows in the range of millions of tokens.
2. Hard to say whether the learning process has actually converged as the optimizations are happening at the prompt level. (Authors mention this in the paper as well).
3. The maximum number of columns for the datasets considered is 37. This might be due to the limitation of model context windows at that time. So this method hasn’t been tested on a large dataset like say 100 columns.
4. The experimental set up is quite novel but optimizing the prompts is not a novel contribution as there have been papers and even frameworks like DSPY who are doing this.
5. Only one model is used for the experiments (GPT 3.5). It will be interesting to see if this method generalizes and scales to other models like Claude, Gemini or open models like llama-3.

**Questions:**

Please address the concerns in weakness

---

> ### Author Response · Authors · 2024-11-19
> **To reviewer hkAL**
>
> We sincerely thank Reviewer hkAL for their thoughtful feedback and for recognizing the contributions and novelty of our study. Below, we address your concerns in detail:
>
> - **Models with longer context windows**: We appreciate the reviewer’s suggestion to explore models with longer context windows that can process all data at once. However, due to data privacy concerns and the risk of data leakage, we cannot benchmark closed-source models like Gemini on our private dataset. Additionally, as demonstrated in our ablation study (Supplementary Figure 5 and Table 7, Lines 391–395),
>
> > Number n of example in in-context few shot learning. Due to the LLM’s limited context length, we implemented a“batches in a batch” method to leverage all training data within these constraints (Section 3.2.1). We varied the number n of examples and found $n = 1$ to be optimal, achieving high
> DCR without compromising MLE (Supplement Section 2.5).
>
> simply increasing the number of in-context samples does not necessarily improve the quality of the generated synthetic data. These findings highlight the nuanced limitations of relying solely on longer context windows for quality enhancement.
>
> - **Convergence.** Thank you for your careful review and for acknowledging our discussion of convergence in the manuscript. We acknowledge that a theoretical convergence guarantee is difficult to obtain as no numerical gradient is available. Instead, we have shown empirical convergence using the optimized parameters such as task instruction (Table 4, Table 10), change of causal structure (Figure 3, Figure 6), and discriminator's accuracy decrease (Table 4, Table 10). In all, we were able to see that the learning process has actually converged at the prompt level (Figure 3, Figure 6, Table 4, Table 10).
>
> - **Ability to deal with high dimensional data.**: We agree with the reviewer that the proposed in-context learning method is constrained by the context length of LLM, limiting its applicability to high-dimensional datasets (e.g., datasets with 100+ columns). However, we believe this limitation can be addressed by switching to LLM with a longer context length, which we can do without substantial technical challenges to cope with. Nevertheless, we agree this is an exciting direction for future investigation and appreciate the reviewer’s perspective.
>
> - **Contribution**:  We appreciate the reviewer’s efforts in exploring related studies. While prompt optimization is not a novel concept and has been applied across various fields, as noted in our manuscript, our contribution is not from just simply utilizing existing prompt optimization. Rather, our contribution is making synthetic data generative models with very low data size. Our novelty is from optimizing causal structure via LLMs and seamlessly linking it to tabular data generative model.
>
> - **Other language models**: We agree that evaluating the generalizability of the framework across different language models would be of great interest. We will enhance our experiments by including other local LLMs or HIPAA-compliant LLM.

---

### Official Review · Reviewer_SHbP · 2024-11-03

**Soundness:** 2
**Presentation:** 2
**Contribution:** 1
**Rating:** 3
**Confidence:** 5

**Summary:**

This paper proposes a method for tabular data synthesis using LLM in-context learning. Specifically, it emulates the architecture
of a Generative Adversarial Network (GAN), where one LLM serves as the generator, one neural network serves as the discriminator, and another LLM serves as the optimizer.  The optimization is conducted on the DAG and instruction part of the generation prompt. Experiments show the proposed method achieves promising results on MLE and DCR metrics.

**Strengths:**

1. This paper is in general well-written and easy to read.
2. This paper reveals a problem with the previous deep generative model, which requires a lot of training data. While LLM in-context learning has the potential to address this problem.

**Weaknesses:**

1. The central part of the optimization phase is to optimize the DAG (Eq 1). The validity of this process relies on the assumption that DAG should play a crucial role in the quality of the generation (otherwise we would not need to optimize it if it does not matter for the generation quality). However, In Lines 398-400, the authors also observe that incorporating the causal structure alone did not significantly improve the MLE compared to a model with only in-context few-shot learning, which challenges the foundation of its method.

2. This paper proposes to prompt another LLM to optimize the parameter $\theta$, based on the history of the previous <score, generation prompt> pair.  I doubt if an LLM is capable of solving this optimization problem, based on two reasons: 1.  the LLM does not have access to the function form of the score.  2. the optimization is extremely hard due to the limited number of previous pairs and the high dimensionality of the parameter $\theta$ (the DAG and task instruction).

3. Continue with Point 2, to see if LLM is able to solve the challenging optimization problem in a meaningful way, more trajectory results need to be shown, similar to Table 4. It is crucial to answer: does each optimization step consistently improve the score? How many steps do we need to converge? What does the trajectory look like for different datasets?

4. **Lack of important baseline**: CLLM [1] is considered out of scope due to its post-hoc data selection. Thus this paper does not compare to CLLM in the experiment.  However, this comparison is crucial since CLLM also only relies on the in-context learning of LLM.  At least a comparison should be done with CLLM without the data selection procedure.

5. **Limited evaluation metric**: This paper only uses two metrics to access the quality of the synthetic data: MLE and DCR, which are too limited. It is strange that the evaluation does not even contain the classification metric the Discriminator used (see Sec 3.2.2). It also lacks many other important metrics used in TabDDPM [2],  column density shape, pair-wise column correlation, and Jensen–Shannon divergence.  This lack of evaluation metrics significantly undermines the convincingness of the proposed method.

[1] Curated LLM: Synergy of LLMs and Data Curation for tabular augmentation in low-data regimes. Seedat et al.
[2] TabDDPM: Modelling Tabular Data with Diffusion Models. Kotelnikov et al.

**Questions:**

See weakness.

---

> ### Author Response · Authors · 2024-11-19
> **To reviewer SHbP**
>
> Thank you for your detailed feedback. I would like to address some of your concerns as list below:
> - **Optimizing DAG**: You are absolutely correct that the core of the proposed model is based on the assumption that by introducing "correct" causal relationships into the prompt can improve the generated data quality. You pointed out that line 398-400 is contradictory. Let us clarify the misunderstanding. In lines 398-400, we claimed that incorporating DAG initialized with a traditional approach (e.g., Hill Climbing heuristic) does not improve the MLE a lot because the data size is too small to learn correct causal relationships. However, in our proposed method, we learned DAG by prior knowledge in LLM, which does improve the MLE. In other words, having incorrect DAG was not helpful, but having correct DAG was helpful.
> - **Prompt optimization**: The concept of transforming a parameter optimization problem into a prompt optimization problem originates from the ICLR 2023 paper [1], “Large Language Models as Optimizers,” as cited in our manuscript (Lines 140–141). We adopted the prompt optimization structure outlined in that paper, which has been demonstrated to be effective across various tasks where an LLM serves as an optimizer given a target score. The focus of our approach is not merely to optimize a score function but to leverage the pre-trained knowledge of the LLM to enrich the context and effectively guide the data generation process.
>
> - **Convergence**: The training trajectory is a well-known challenge in the GAN field. During training, we observed that the discriminator’s accuracy score initially decreased over the first few updates and then stabilized, fluctuating around 0.5. However, simply monitoring the discriminator’s accuracy is insufficient to confirm convergence, as the model may still suffer from issues like mode collapse. In the original submission, we addressed this by including additional training trajectory details in Supplementary Table 4 and Table 10, which provide insights into the training progress from both a scoring and prompt perspective. Additionally, we acknowledged in the limitations section that the proposed method incurs high computational costs due to iterating over the entire dataset. To mitigate these challenges, we implemented several strategies. These include using smaller batches during “gradient” descent to allow for more frequent updates, thereby improving convergence speed compared to processing the entire dataset at once. Furthermore, we employed a simpler model with incremental updates, which helps stabilize the discriminator during training. These measures collectively enhance the robustness and practicality of our method.
>
> - **Lack of important baseline**: We agree with the reviewer that CLLM is an important baseline to consider, as it also utilizes the in-context learning method to address low-data regimes. In fact, we have already included results of CLLM in the original submission. The prompt style we used is similar to that of CLLM, and we did not perform any additional data curation for the downstream task, as suggested.
>
> > Lines 398–399: We compared the full model, which includes both components, to a version without them, similar to CLLM (31) without post-processing data selection (Table 3).
>
> | Dataset (Metric) | Few-shot | Few-shot+Causal | Few-shot+Causal+Opt (ours) |
> |------------------|----------|-----------------|----------------------------|
> | **Adult (F1)**   | 0.7550 ± 0.0454 | 0.7503 ± 0.0393 | **0.7892 ± 0.0358**      |
> | **Asia (F1)**    | 0.2335 ± 0.0000 | 0.2756 ± 0.2842 | **0.8282 ± 0.0041**      |
> | **Insurance (R²)** | 0.6821 ± 0.0193 | 0.6718 ± 0.0916 | **0.7152 ± 0.0447**      |
> | **ATACH (R²)**   | 0.1581 ± 0.0850 | 0.1326 ± 0.0637 | **0.2726 ± 0.0707**      |
> | **ERICH (R²)**   | -0.0647 ± 0.0701 | **0.0281 ± 0.0424** | -0.0253 ± 0.0671    |
>
> - **Missing evaluation metric** We sincerely thank the reviewer for their thoughtful suggestions regarding evaluation metrics. We fully acknowledge the importance of robust metrics in assessing the quality of synthetic data. The discriminator's accuracy you mentioned is in Table 6 due to limited space. We followed the evaluation framework in [6].  We will include all those important metrics in the revised version. We appreciate your suggestions.
>
> [6]  "Language models are realistic tabular data generators. In The Eleventh International Conference on Learning Representations 2023."

---

> ### Comment · Reviewer_SHbP · 2024-11-26
>
> Sorry for my late response. I appreciate the authors' effort in addressing my concerns. While some of my initial questions have been clarified, some issues remain:
>
> 1.  I still found the effect of incorporating DAG into the prompt unclear, and I still doubt whether LLM can find the correct DAG.
>
> >  In lines 398-400, we claimed that incorporating DAG initialized with a traditional approach (e.g., Hill Climbing heuristic) does not improve the MLE a lot because the data size is too small to learn correct causal relationships.
>
> I acknowledge that this paper studies the low data resource setting. However, I believe it is necessary to examine if the LLM prompt optimization procedure can actually find the correct DAG. To do this, we can use the full training set and apply traditional methods [1] to estimate the DAG from data.  Then we can compare the DAG produced by traditional methods and the one optimized by the LLM.  Does the LLM optimize the DAG to converge to the correct one?
>
> 2. About convergence.
>
> > However, simply monitoring the discriminator’s accuracy is insufficient to confirm convergence, as the model may still suffer from issues like mode collapse.
>
> The rebuttal acknowledges that discriminator accuracy alone is insufficient for confirming convergence due to potential issues like mode collapse. However, this raises a critical question: what are the specific criteria used to determine when training should stop? This needs to be clearly defined and justified.
>
> 3. About evaluation metrics.
>
> >The discriminator's accuracy you mentioned is in Table 6 due to limited space.
>
> Table 6 only contains results for one dataset, and it lacks comparison with other deep generative model baselines. Also, this table is not referred to in the main context, and I cannot find any text describing the details of it.
>
> In summary, I think this paper needs significant revision to improve its clarity, motivation, and convinceness, thus I will keep my score.
>
> Reference: [1] Ankan, Ankur, Abinash, Panda. "pgmpy: Probabilistic Graphical Models using Python." Proceedings of the Python in Science Conference. SciPy, 2015.

---

### Official Review · Reviewer_iZoG · 2024-11-05

**Soundness:** 3
**Presentation:** 3
**Contribution:** 3
**Rating:** 5
**Confidence:** 4

**Summary:**

In this paper, the authors propose a novel model that generates synthetic tabular data via a proposed multi-agent LLMs framework. The proposed method aims to handle the issues of limited training data size in healthcare. The proposed framework ICL and does not require fine-tuning on LLM and the ICL examples are obtained by a "multi-agent LLM AS GAN" model.

**Strengths:**

Prons:

1. The experimental setting seems solid as they conduct the experiments on 5 different datasets.

2. The idea of combining LLM and GAN makes sense.

3. The proposed method is well-generalized, simple, and can be applied to many tasks.

**Weaknesses:**

Cons:

1. From the experimental results shown in Tables 2 & 3, the proposed model cannot always obtain the SOTA performance (In table 2 & 3).

2. The main idea is to combine LLM (with ICL ) and GAN and the idea is somehow straightforward. To highlight the technical contribution of this paper, the authors should make it clear the technical difficulty of combining ICL-LLM and GAN and clarify the technical contribution (novelty) of this proposed method.

3. This paper aims to handle the training on small datasets. To achieve the model training on small dataset, the authors selected sets of samples from the whole (large) datasets to compose small datasets (N=100, 200, ..., 800). However, the dataset spliting is conducted by the authors instead of using some standard benchmarks. Is it possible to directly use benchmarks with small datasets? It means the authors need not split the dataset by itself and use the small dataset being used in other papers. By doing so, the comparsion will be more fair and solid.

**Questions:**

None.

---

> ### Author Response · Authors · 2024-11-19
> **To reviewer iZoG**
>
> We sincerely thank Reviewer iZoG for thoroughly reading our paper and providing thoughtful feedback. Below, we address your three key concerns:
> - **Performance concern.** We appreciate the reviewer’s careful examination of our results. While we agree that our proposed model does not consistently achieve state-of-the-art (SOTA) performance across all scenarios, the primary goal of our work is to address data scarcity in low-data regimes (e.g., N=100, 200). Our intention is not to claim that our model outperforms all other data-driven methods in general settings. The results for moderate (N=400) and larger datasets (N=800) are included to demonstrate that our model remains competitive, achieving 1st or 2nd place across all datasets, as shown in Table 2. This indicates that our method performs robustly even when sufficient training data is available.
>
> - **Highlight of contribution.** We thank the reviewer for acknowledging the technical contribution of our proposed method. Actually, the technical challenge we addressed in our work was generating synthetic data when the data size was very small. Integrating ICL-LLM and GAN was our solution. We do believe that lines 084-087 have highlighted the novelty of our proposed solution. Regarding technical challenges when integrating LLM and GAN, we encountered and addressed several technical challenges, such as limited context length (lines 193-197), intractability of discriminator over iterations (lines 302-305), and degrading discriminator's performance for convergence (line 245-249).
>
> - **Standard benchmark.** We appreciate the suggestion to incorporate standard benchmarks. To the best of our knowledge, this is the first paper to benchmark synthetic data generation across datasets of varying sizes. While incorporating standard benchmarks could make this paper more solid, it also introduces potential biases, such as data leakage from LLM training data. To address this, we included evaluations on two private datasets, Lines 306–310:
>
> >"Data. Our benchmarks include several datasets from various domains: three public datasets (Adult(4),
> Medical Insurance(1), Asia(30)), and two private medical datasets (ATACH2, ERICH) (22). To
> ensure fair comparison without memorization concerns of LLM (e.g., public datasets are in the
> training corpus of LLM), private datasets were included. Details are in Supplement Table 5."
>
> To alleviate concerns, we will make all subsampled public datasets available for future benchmarking, ensuring transparency and reproducibility.

---

### Comment · Area_Chair_zGwj · 2024-11-25
**Please check author responses and participate in the discussion**

Dear Reviewers,

Thank you for your efforts and contribution to ICLR! The authors have posted their responses to your original comments. Since only less than two days are left for the reviewer-author discussion, your help and prompt responses are important. Please actively check the authors' responses and actively participate in the discussion.

Thanks!

Best regards,

Your AC

---

### Meta-Review · Area_Chair_zGwj · 2024-12-23

**Metareview:**

## Summary:
The paper introduces a novel framework for generating synthetic tabular data using large language models (LLMs) inspired by Generative Adversarial Networks (GANs). The proposed approach leverages the in-context learning capabilities of LLMs without requiring fine-tuning, improving data generation quality in scenarios with limited training data, particularly in healthcare applications. It adopts an architecture of two LLMs + a discriminator (a classifier of real vs. synthetic data), where one LLM acts as the data generator and another as the optimizer of the prompt for the generator based on the feedback from the discriminator. The optimization focuses on the DAG and instruction part of the generation prompt, refining causal relationships and task instructions to enhance the generated data quality. Experimental results demonstrate the effectiveness of the method in surpassing state-of-the-art models in generating high-quality synthetic data for downstream tasks while maintaining data privacy.

## Strengths:
1. The paper is well written with the ideas and main results presented.
1. Applying the idea of GAN to improve the generation of synthetic data by ICL on LLMs is novel and the explicit optimization process leads to better explainability.
1. The method does not require model finetuning. Instead, it provides feedback signals to the generator via the LLM optimizer, and refines prompts to improve data generation quality without the need for fine-tuning.

## Weaknesses:
1. Non-standard datasets and splitting in the experiments. While the data leakage might introduce possible bias, results on the standard benchmarks should be also reported for a better comparison with previously reported results.
1. The scalability of this method is undermined by the expensive cost of an optimization loop involving the inference on two powerful LLMs and the discriminator per iteration.
1. The method does not always achieve better performance than the baselines under the same budgets. Though privacy protection is another advantage as reflected by the reported DCR, it is not clear what range of DCR may lead to high risks of data leakage according to existing theories of privacy leakage.
1. Several important synthetic data baselines and ablation studies raised by the reviewers have not been examined in the experiments or the discussion. Simple baselines applying ICL to the latest close-source LLMs need to be compared. Due to the complex architecture of the proposed pipeline, these comparisons are critical to justify whether the proposed GAN strategy is the main reason for the advantages.
1. Since "LLM as an optimizer" and "ICL on LLMs generates synthetic data" have been widely studied in the literature, the original novelty of this paper is limited to multi-round optimization.


## Decision:
The authors provided further clarifications and additional experimental results in the rebuttal, as requested by the reviewers. One reviewer participated in the author-reviewer discussion phase. The meta-reviewer carefully checked the responses and the new experiments. Despite the improvement brought by them, the reported experiments are not sufficient or clear for a thorough examination. The problem setup requires more justifications, while the advantages of the proposed method of balancing performance and privacy are currently vague. Since the paper has not received positive ratings from the reviewers, and based on the above summary, the meta-reviewer does not recommend the paper for publication on ICLR. That being said, the idea is novel and the revised draft is encouraged to be submitted to a new venue.

**Additional Comments On Reviewer Discussion:**

The authors provided further clarifications and additional experimental results in the rebuttal, as requested by the reviewers. One reviewer participated in the author-reviewer discussion phase. The meta-reviewer carefully checked the responses and the new experiments. Despite the improvement brought by them, the reported experiments are not sufficient or clear for a thorough examination. The problem setup requires more justifications, while the advantages of the proposed method of balancing performance and privacy are currently vague. Since the paper has not received positive ratings from the reviewers, and based on the above summary, the meta-reviewer does not recommend the paper for publication on ICLR. That being said, the idea is novel and the revised draft is encouraged to be submitted to a new venue.

---

### Decision · Program_Chairs · 2025-01-22

Reject